# Predicting Bordeaux red wine origins and vintages from raw gas chromatograms

Michael Schartner [1], Jeff M. Beck[2], Justine Laboyrie[3], Laurent Riquier[3], Stephanie Marchand[3✉] & Alexandre Pouget [4✉]

Connecting chemical properties to various wine characteristics is of great interest to the science of olfaction as well as the wine industry. We explored whether Bordeaux wine chemical identities and vintages (harvest year) can be inferred from a common and affordable chemical analysis, namely, a combination of gas chromatography (GC) and electron ionization mass spectrometry. Using 12 vintages (within the 1990–2007 range) from 7 estates of the Bordeaux region, we report that, remarkably, nonlinear dimensionality reduction techniques applied to raw gas chromatograms recover the geography of the Bordeaux region. Using machine learning, we found that we can not only recover the estate perfectly from gas chromatograms, but also the vintage with up to 50% accuracy. Interestingly, we observed that the entire chromatogram is informative with respect to geographic location and age, thus suggesting that the chemical identity of a wine is not defined by just a few molecules but is distributed over a large chemical spectrum. This study demonstrates the remarkable potential of GC analysis to explore fundamental questions about the origin and age of wine.

[1] Center for the Unknown, Champalimaud Institute, Lisbon, Portugal. [2] Duke University, Durham, NC, USA. [3] Université de Bordeaux, ISVV, INRAE, UMR 1366 OENOLOGIE, 33140 Villenave d'Ornon, France. [4] Département des neurosciences fondamentales, Université de Genève, Genève, Suisse. ✉email: stephanie.marchand-marion@u-bordeaux.fr; alex.pouget@gmail.com

dentifying and characterizing the origin of wines on the basis of their chemical content is a challenging yet fundamental problem in wine science. Wines are shaped by multiple factors such as the soil, the climate[1,2], the varietals, the microbiology and the wine-maker's practices. If we are to understand how these factors influence the taste of a wine, we need to uncover what part of the chemical composition determines its quality, origin, and typicity. Wine typicity and authenticity are indeed at the center of the wine industry's preoccupations[3–7].

One approach to address these questions consists in measuring the concentration of specific targeted molecules that are thought to be particularly informative with regard to wine origin and flavor. This approach has led to the identification of several key compounds but it is akin to finding a needle in a haystack[8–14]. Unlike industrial beverages, wines are complex mixtures of molecules and their taste often depends on molecules present only in remarkably small concentrations. Also, it is quite possible that the chemical typicity of wine is not defined by the concentration of a handful of molecules but depends instead on the overall pattern of concentrations over a wide range of molecules, possibly in a nonlinear way.

Several groups have indeed abandoned the targeted approach in favor of a global perspective. They used statistical tools from machine learning (ML) to analyze the output of broad-spectrum chemical analysis. This approach has been applied for instance to wine classification using ICP MS[15], nuclear magnetic resonance[16], RP-HPLC/DAD[5] and UV-spectroscopy[17] or to wine region classification using GC/QTOFMS[18], isotopic ratio[19], absorbance-transmission and fluorescence excitation-emission matrix (A-TEEM) or climate data[20]. Other studies have also looked at sensorial properties and aroma profiles using gas chromatography (GC)[21] and wine quality using global chemical measurements (alcoholic contents or acidity as examples)[22] and the emergence of oxidative markers during aging with GC[23].

Here we apply (ML) techniques to raw chromatograms obtained with GC. GC is a popular type of analysis for wine, which has led to seminal discoveries, but which is typically used within the targeted molecule approach, i.e., as a tool to identify molecules of interest. Yet, GC is a lot more amenable to global analysis because it can reveal the presence of a much wider range of molecules than, say, UV-spectroscopy. The problem, however, is to determine what class of chemicals to target with GC, since GC can be tuned to reveal specific families of molecules (e.g., esters) depending on the type of filter applied to the output of the chromatography column. Moreover, it is unclear which part of the chromatogram to focus on. Typically, researchers integrate peaks of the chromatograms to quantify the concentration of specific molecules, thus discarding the rest of the chromatograms. Using the raw, unprocessed chromatograms could in principle lead to better results but this requires using techniques that can automatically determine which parts of the chromatograms are most informative for a particular classification problem. This is precisely where ML can help since ML algorithms can automatically find the most informative parts of a chromatogram.

Accordingly, we used a variety of ML techniques, including nonlinear dimensionality reduction, linear and nonlinear classifiers, and regression models to predict several features of wines from the Bordeaux regions from chromatograms. We focused in particular on the identity of estates and vintages. Our results indicate that raw chromatograms are highly informative about terroir and estate identity for the Bordeaux estates studied here. We also found that integrating peaks to estimate the concentration of specific chemical compounds leads to lower performance on all tasks. Finally, whether we use the raw chromatograms or a quantification table of specific compounds, we observed that wine chemical identity is defined by a large chemical spectrum rather than a few specific molecules.

## Results

Our results are based on three types of GC methods which, for simplicity, we refer to as esters, oak, and off-flavor (offFla) in the rest of the paper (see Fig. S1 for examples). However, these chromatograms correspond to different extraction strategies (see Methods), which are not exclusively sensitive to the targeted molecules from which the method names arise. It is, therefore important to keep in mind that the resulting chromatograms are not simply reflecting the wine content for these three selections of molecules. Also, we emphasize that this data set was not collected specifically for the present study but had been used previously for a different purpose[24]. Nonetheless, given its diversity, we thought that this data set has the potential of being informative about the estate and vintage of wine.

**Recovering terroir through dimensionality reduction**. We first applied nonlinear dimensionality reduction to the gas chromatograms, allowing us to visualize the distribution of the wines in 2-D. This first analysis was based on the concatenation of the three types of chromatograms ("esters", "oak", and "off-Fla", see Methods) into a single meta-chromatogram per wine. Figure 1a, b shows the results for two different clustering techniques, t-distributed stochastic neighbor embedding (t-SNE) and UMAP, which project the data in such a way that wines with similar chromatograms remain close to one another in the projected 2-D space.

Several organizational principles emerged from these projections. First, wines from the same estate (or 'chateau', as they are known in the Bordeaux region) tend to form distinct clusters regrouping all vintages, with very few outliers (e.g., A-1990 and C-1990). This suggests that the chromatograms reveal specific features of each estate independently of the vintage. Second, two large clusters are clearly visible in both Fig. 1a, b, regrouping the A, C and B in one cluster and D, E, F, and G in another. Within the second cluster, G and F tend to lie on one side, closer to the (A, C, B) cluster while D and E lie on the other side of the cluster. Strikingly, the spatial configuration of the projection reflects the geography of the Bordeaux region (Fig. 1c) in the following way. Wines (A, C, B) are all located on the right bank of the Garonne river in the Libourne region, while (G, F, E, D) are all left-bank estates from the Medoc. Furthermore, D and E are located in the northern part of Medoc, while G and F are further south, next to the city of Bordeaux, in between (A, C, B) and (E, D), as in Fig. 1a, b. In other words, spatial relations between wines in 2-D-embedded space are similar to the corresponding wine estates' geographical relations.

We note that this is not true within the right bank estates, which are not spatially arranged in a way consistent with their relative geographical locations, but which also happen to be particularly nearby (within 7 km, versus 40 km between C and G, the two closest right and left bank estates). We also tried principal component analysis (PCA), a linear dimensionality reduction technique, and found that the clusters were not as clear, though the right-left bank distinction was still evident (Fig. S2). Further note that cluster spread and orientation depend on cluster algorithm parameters such as the random seed, though the above-described spatial relations remain invariant.

Next, we applied this analysis to the three types of chromatograms separately (Fig. S2). While right and left bank estates tend to cluster together for all three chromatograms, this separation is particularly clear with the "offFla" data. However, the "offFla" (SBSE-GC/MS) do not show clear estate clusters while these clusters

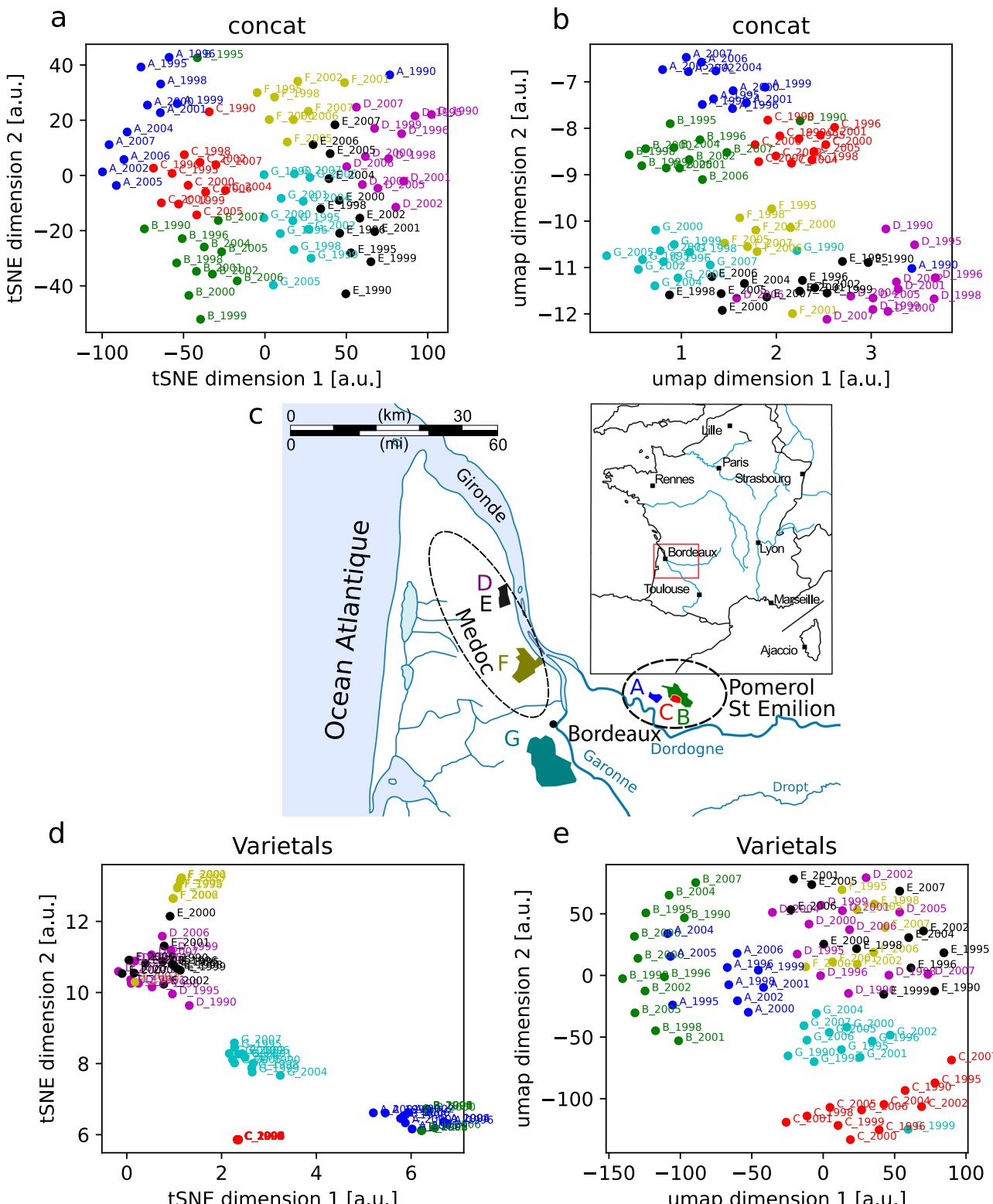

are clearly visible for the oak (liquid extract) and esters (SPME-GC/MS). As to the north-south axis of the left bank, it is only observed in oak, where we can see that the estates G and F stand in between the right bank estates and the estates T and F, just as we observed for the concatenated chromatograms.

We hypothesized that the following factors are likely to contribute to these results. First, these estates use different blends of four varietals: cabernet-sauvignon, cabernet franc, merlot, and petit verdot. The percentage of these varietals varies across estates and across vintages (Table S1). To evaluate whether this variability in blends is sufficient to explain our results, we applied the same dimensionality reduction technique to the percentages in the blend (reducing 4 dimensions to 2). Figure 1d, e shows the resulting plots. While three estate clusters (C, G, F) are clearly separated, other estates are now indistinguishable (A and B, and E and D). Moreover, the distinction between right vs left bank is

**Fig. 1 Dimensionality reduction of chromatograms reflects the geography of the Bordeaux region. a** t-SNE plot of the 80 concatenated chromatograms with the first two embedding dimensions. Colors correspond to different estates while vintages appear next to each data point. The resulting map recapitulates the geography of the Bordeaux region up to a rotation, but note that the overall orientation of the t-SNE projection is arbitrary. Wines from the same estate but different vintages tend to cluster together, with little overlap between clusters. Right bank estates (A, C and B from Pomerol and St-Emilion) and left bank estates (F, G, D, and E from Medoc) also tend to cluster together. Moreover, the left bank estates are organized along a north-south axis, E and D being the furthest north, while G and F are closer to Bordeaux. **b** Same as (**a**) but with the UMAP algorithm. **c** The 7 estates in our data set are shown in the same colors as in (**a**), coming from the two regions in South-West France (inset), highlighted by the ellipses on the right (Pomerol and St-Emilion) and left (Medoc) bank of the Garonne river. The clusters follow the same general organization as for t-SNE. **d** t-SNE applied to varietal percentages. **e** same as (**d**) but with UMAP. In (**e**), the distinction between right and left bank estates is less clear, and in both (**d**) and (**e**), there is no north-south axis on the left bank and some estates (A and B, E and D) are no longer distinguishable. This suggests that the blend is not the sole contributor to the map obtained with GC.

less marked (in particular for UMAP for which C tends to be closer to the left bank estates). Finally, the south-north axis in the Medoc region is no longer present (G and F are not next to each other and do not stand in between the E-D estates and the right bank estate). This indicates that while the blend plays a partial role, it is not the only contributing factor. Other factors are likely to contribute such as the composition of the soil, vine and climate, modulated by the wine-making practices of each estate.

**Estate and vintage identification**. Given the clear estate clusters generated by t-SNE and UMAP, one would predict that it should be possible to identify estates with high accuracy independently of vintage from the chromatograms. Conversely, we expected that vintage identification, independently of the estate, might be more challenging given that vintages do not seem to cluster together (Figs. 1a, b and S2).

This is indeed what we found, using LDA and logistic regression (LR). Fig. 2 shows the histogram of test performance (i.e., generalization to unseen data) across multiple splits and for the three chromatograms independently or for the concatenated chromatograms. On average, the best performance was obtained with LDA applied to the concatenated chromatogram, leading to 99% correct estate classification. Interestingly, similar performance was observed using only the oak chromatogram, or the ester chromatogram, while the offFla chromatogram led to worse performance (87% correct).

We used the same technique to decode vintages. In line with the results of t-SNE and UMAP, and in sharp contrast to estate decoding, we obtained relatively low decoding performance on vintages, with a best performance of 27% correct with LDA applied to the oak chromatogram. While small, this is still well above chance performance of 8% ($p < 0.001$, Bonferroni corrected). Decoding vintages from the esters and offFla chromatograms led to near-chance performance.

**Detailed analysis of chromatograms**. Wine chemical identity could be defined primarily by the concentration of a handful of molecules or, conversely, by the overall pattern of concentration over a large range of molecules. To explore this issue, we looked for the regions of the chromatograms that most contribute to classification performance. We did so by first binning the chromatogram into 50 bins and then removing one by one the least informative bins (a procedure we coined 'survival of the fittest'). This procedure revealed that decoding performance remains stable (and sometimes even increases due to overfitting) after removing about 45 bins (Fig. 3a). This was observed for both estate and vintage classification. Figure 3a (lower panel) shows the location of the remaining 5 bins on the chromatogram. Interestingly, the most informative bins do not necessarily line up with the largest peaks of the chromatogram suggesting that classification performance is partly driven by molecules with very low average concentration ($\mu g.L^{-1}$ or less) (Fig. S1 shows

examples of normalized chromatograms, revealing the presence of multiple small peaks).

This also shows that only a small fraction of the chromatogram is sufficient to reach asymptotic classification performance. We verified this by training our classifier on the concatenation of the N most informative bins for each of the three chromatograms, where N was systematically varied from 1 to 4, for a total of 3xN bins. We found that keeping 2–3 bins per chromatogram is sufficient to match, and in some cases outperform, the classifier trained on the full concatenated chromatograms. Indeed, the LDA estate classifier performed perfectly, 100% correct, when trained on the concatenation of the top 3 bins for each chromatogram, compared to 99% correct for the full concatenated chromatogram (Fig. S3).

In the case of vintages, classification reached a performance of 34% correct when training on the concatenation of the top 3 bins for each chromatogram, compared to 27% for the full concatenated chromatograms. Performance reached up to 50% correct when using only the most informative bins of the oak chromatogram (Fig. 3c). We also observed that some vintages are easier to decode than others, with 2007 being clearly the easiest (Fig. S4).

These results can be slightly misleading in that they suggest that there are just a few informative bins in each chromatogram. It is possible instead that most bins are informative but also happen to be highly redundant. The PCA analysis of the chromatograms already points in that direction since 90% of the variance in our data set is explained by only 20 dimensions (Fig. S5). In order to explore this issue further, we trained classifiers on each bin separately. The red histograms in Fig. 3b show the estate classification performance for the individual bins in all three chromatograms. Surprisingly, while some bins are more informative than others, the vast majority of the bins lead to similar estate classification performance, well above chance (14%), indicating that many parts of the chromatograms contain information about estate identity and vintage (Fig. 3d, chance level at 8%).

This conclusion is consistent with the profile of the weights used by the LDA classifier (Fig. S6). If only a few molecules mattered, one would expect the weight pattern to show a few prominent peaks, with smaller weights between the peaks. Instead, we see that the weights are homogeneously large throughout the chromatograms. Moreover, we ran an analysis in which we removed one by one the most informative bins while tracking classification performance (the opposite of the 'survival of the fittest' procedure above). We found that performance declines gradually, and drops sharply only when there is about 20% of the least informative bins left, thus revealing that the most informative bins do not play any specific role in encoding the chemical identity of the estate (Fig. S7). Altogether, these results strongly suggest that the chemical identity of an estate does not rely on the concentration of a few molecules but rather on the whole chemical spectrum.

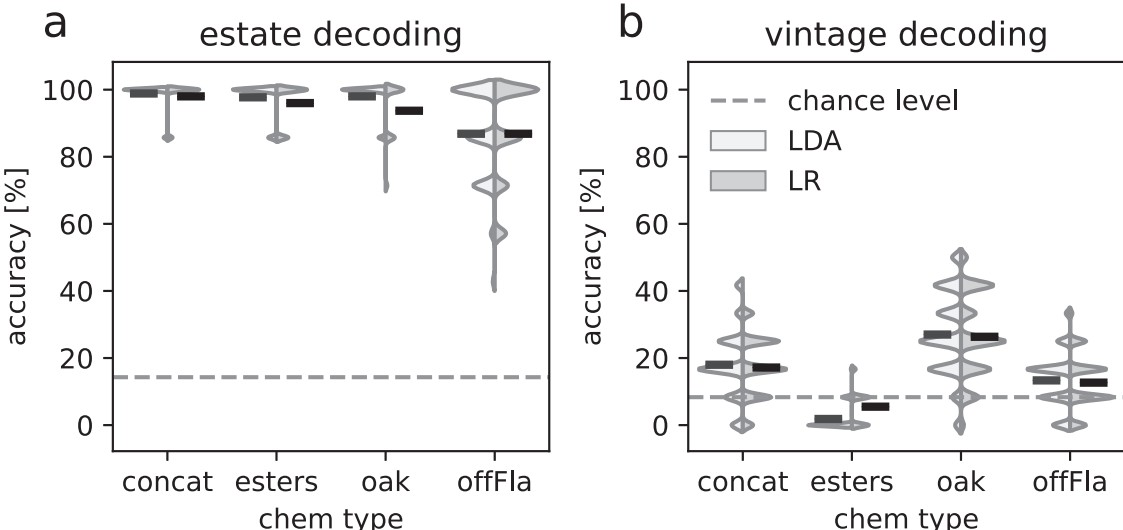

**Fig. 2 Supervised vintage and estate decoding. a** Performance histograms for decoding estate identity using two classifiers - linear discriminant analysis (LDA) and logistic regression (LR). The horizontal black line in each histogram indicates mean performance across data splits. Chance performance (14%) is shown by the dashed line. Note that the histograms are smoothed, which is why they can go beyond 100. Each peak corresponds to one of the possible accuracies per run. The best average decoding accuracy was 99% correct for LDA and concat (all three types of chromatograms concatenated together), though comparable results were observed for the oak and ester chromatograms. Performance was markedly weaker for the offFla chromatogram (87% for both, LDA and LR). **b** Same as in (**a**) but for decoding vintages. Decoding performance was smaller overall than for estate identity, however performance for LDA and LR applied to the oak chromatogram was significantly above chance ($p < 0.001$, 27% correct versus 8% for chance) and, to a lesser extent, for offFla chromatogram ($p = 0.0012$). Concatenating chromatograms did not lead to stronger performance.

**Chemical-compounds based analysis**. To further explore whether wine chemical identity relies on a large ensemble of molecules or a small subset, we manually extracted the area under the peak for 32 chemical compounds from the chromatograms, converted these area measurements into concentrations, and repeated the same set of analyses (16 compounds from Ester, 13 from Oak and three from offFla, see Methods for a complete list of compounds, information on internal standards and calibration strategies). In these new analyses, each wine is now characterized by a 32-dimensional vector of compound concentrations.

Dimensionality reduction using either tSNE or UMAP reveals estate-specific clusters though with significantly more overlap than was observed for the maps from the raw chromatograms (Fig. 4a, b). Moreover, the right/left bank distinction is less clear, particularly in the case of tSNE which does not group the right bank wines (A, B, C) together.

With regard to wine estate classification, we found the performance was markedly reduced (Fig. 4c). For instance, performance for logistic regression using oak decreased from 95% correct with the chromatograms to 78% with the oak compounds. Likewise, performance decreased from 98% to 75% for ester and 85% to 27% for offFla. This suggests that many of the smaller peaks in the chromatograms strongly contribute to the estate identity. This also shows that the traditional approach of extracting the concentration of targeted compounds is not as good as working with the raw chromatograms, and has the disadvantage of requiring extensive manual pre-processing.

The results for vintages on the other hand were mixed (Fig. 4d). Oak performance for logistic regression decreased from 27% with the chromatogram to 23% with the compounds. However, we observed the reverse for ester, for which performance increased from 7% to 23%. Still in all cases, performance remained below the best vintage performance of 27%, which was obtained with the oak chromatograms.

We saw previously that vintage classification performance could be improved to 50% for the chromatogram when using our survival of the fittest procedure. The same approach applied to the compound concentrations also improves performance but only to 37% correct (Fig. S10). Therefore, once again, using compounds does not improve vintage classification performance and, in this case, even results in lower performance.

One advantage of working with compound concentrations is that we can measure the influence of any compound on classification performance by examining the classification weights. Interestingly, we found that the weights for estate classification tend to have similar values across all compounds, indicating that most chemical compounds contribute to the identity of the estate (Fig. S11). We also trained classifiers based on the concentration of single compounds, similar to our analysis based on single chromatogram bins (Fig. 3b). Some compounds lead to up to 40% correct classification on their own (such as acetosyringaldehyde, ethylbutanoate-C4C2 and ethylhexanoate-C6C2) but, remarkably, estate identity can be significantly decoded from the concentration of any of the 32 compounds (Fig. S12). This brings further support to our claim that estates are not defined by the presence of a few specific chemicals but instead by the overall chemical spectrum

## Discussion

Gas chromatography has a long history in wine science, dating back to the 1980s, but our results revealed that this analysis might be more powerful than had been suspected hitherto. The fact that nonlinear dimensionality reduction techniques recovered the geography of the Bordeaux wine regions showed in particular that the raw gas chromatograms provide a chemical signature of terroir, i.e., a combination of the soil, rootstock, varietal, location, blend, and winemaking practices. In addition, for the seven estates we considered, estate identity can be predicted perfectly from chromatograms, independent of vintage, while vintage could be recovered with 27% accuracy, and up to 50% correct when targeting a specific part of the chromatograms.

Other groups have recently started to use ML approaches to classify estates or vintages from chemical measures. In particular, Ranaweera et al.[25] have reported nearly perfect performance

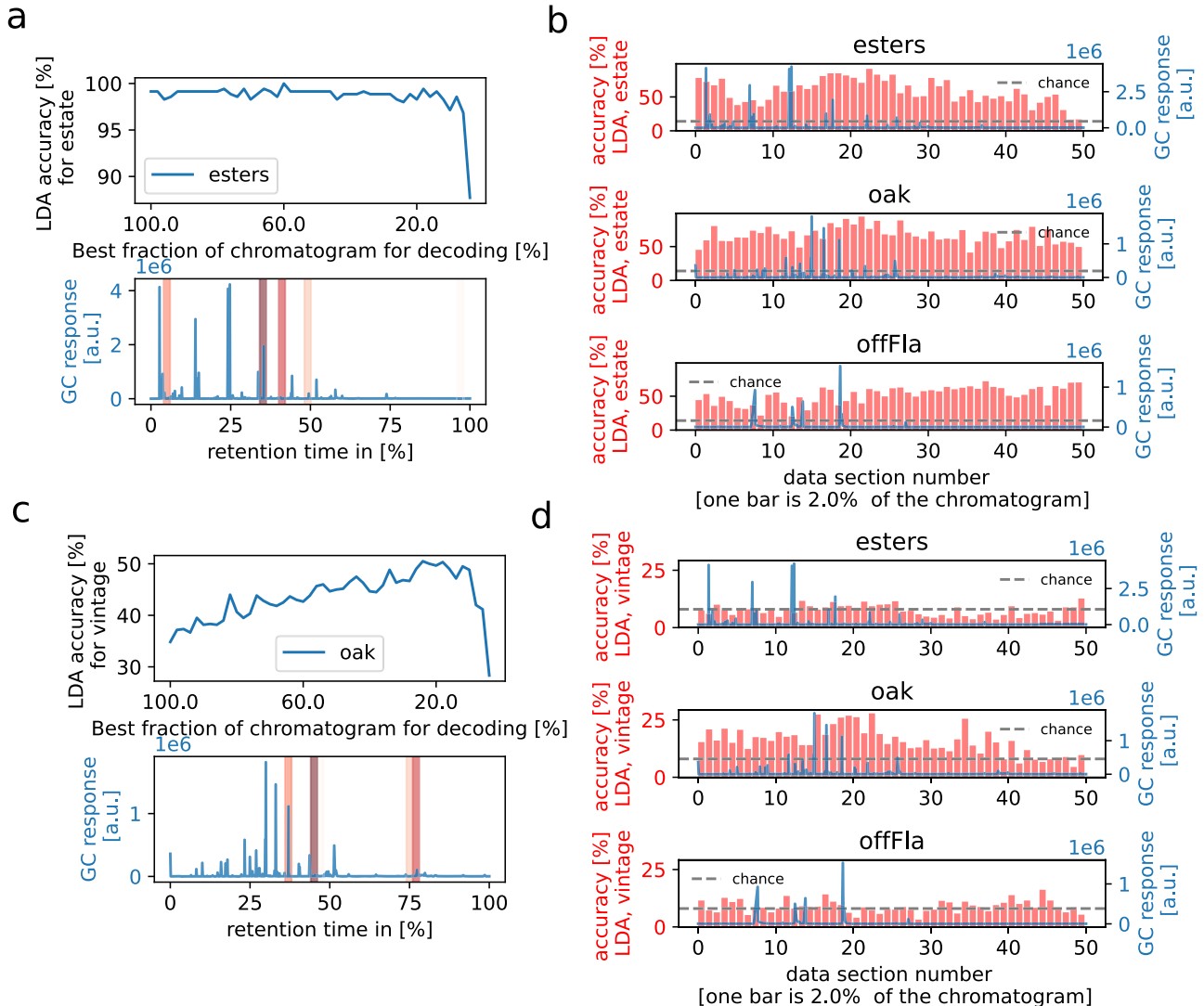

**Fig. 3 Identifying most estate-informative sections of the chromatograms. a** A "survival of the fittest" algorithm was applied, removing the 2% bins of the ester chromatogram that had the least effect on estate decoding accuracy, before removing the next bin one by one, until the last bin was left. Importantly, the same decoding accuracy was achieved with the best 10% of the total data than with the complete chromatogram, showing that these sections have all the estate information. The top panel shows the decoding accuracy as a function of the fraction of the data with the best decoding accuracy. The lower panel shows the five most important sections (red) on top of an example ester chromatogram (blue). The red color darkness indicates their rank in the survival algorithm, darker bins being more informative. **b** Estate decoding accuracy per data bin (red bars) with an overlaid example chromatogram (blue), for each chromatogram type. After dividing the chromatogram into 50 equal bins, estate decoding was performed using only single bins with LDA as in Fig. 2. Test decoding accuracy is shown in red for each bin, fluctuating fairly continuously across section locations in the chromatogram with most having above-chance (0.14) decoding accuracy. This indicates that estate chemical identity is not defined by just a few bins of the chromatogram but is distributed throughout. The fact that the decoding performance only requires 5 bins (**a**) suggests that the information across bins is highly redundant. Similar results were obtained for oak and offFla (Figs. S8, S9). **c, d** show the results of the same analyses performed for vintage decoding. Note that the reduction of the oak chromatogram led to a 20% increase in performance, indicating that our decoder was subject to overfitting when applied to the whole chromatogram. Decoding performance from individual bins is lower than for estate decoding yet still clearly above chance for most segments, again suggesting that vintage information is distributed throughout the chromatogram and that there is a high level of redundancy across bins.

in classifying three vintages and five subregions of Barossa valley shiraz based on absorbance-transmission and fluorescence excitation-emission matrix (A-TEEM). It is not clear whether this technique could identify estates independent of vintages as we report here, but the fact that their approach could clearly distinguish vintages suggests that a combination of A-TEEM and GC might improve joint classification of estate and vintage. The study of Li et al.[18] is also particularly relevant for our approach given that they analyzed Australian shiraz with GC, and more specifically GC/qTOF-MS. However, they used a set of wines that are separated by up to 1500 km and included only two vintages.

Moreover, they did not use raw chromatograms but manually extracted features. In that respect, our analysis is simpler since it is based on raw chromatograms. It is also cheaper as GC is considerably less expensive than GC/qTOF-MS. Nonetheless, this study complements ours in that it confirms the potential of using GC for wine classification, even for single-varietal wines.

Our results also suggest that wine's chemical identity is not defined by the concentration of just a handful of molecules. Indeed, we have found that wine classification does not depend critically on specific regions of the chromatograms and, therefore, on the concentration of specific molecules. Instead, most bins of

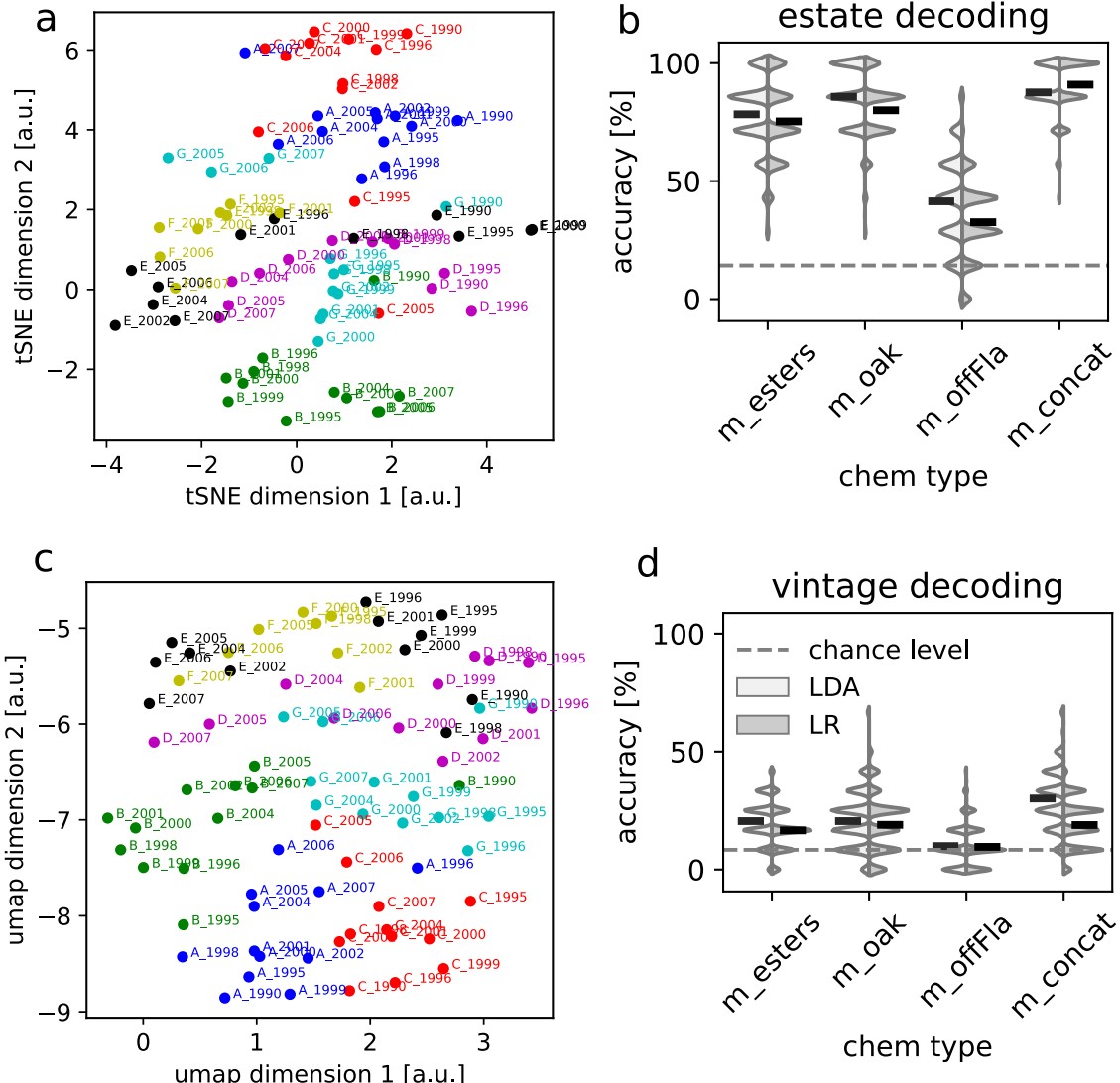

**Fig. 4 Estate and vintage decoding from compounds. a, b** Dimensionality reduction of 32 compounds via tSNE and UMAP. The estate clusters are not as well marked as with the raw chromatograms, and the right bank wines (A, B, C) are not clearly separated from the left bank ones (D, E, F, G), particularly in the tSNE case (Fig. 1). **c** Performance histograms for decoding estate identity from subsets of compounds, using two classifiers: Linear discriminant analysis (LDA) and logistic regression (LR), as in Fig. 2. The horizontal black line in each histogram indicates mean performance across data splits. Chance performance (14%) is shown by the dashed line. The best average decoding accuracy was 91% correct for LR and m_concat (all three types of compounds concatenated together, 32 dimensions), though comparable results were observed for the oak and ester compounds. Performance was markedly weaker for the off-Flavor (offFla) compounds (41% for LDA). Overall performance is worse than with the raw chromatograms (Fig. 2). **d** Same as in (**c**) but for decoding vintages. Decoding performance was smaller overall than for estate identity, however performance for LDA and LR was significantly above chance ($p < 0.001$, 8% for chance) for all but offFla. Concatenating the three compound types into m_concat leads to slightly stronger performance.

the chromatograms contain information about estate identity. However, classification does not require the entire chromatograms for near-perfect performance; 10–20% is typically sufficient. This indicates that the information about estate identity is distributed across the whole chromatogram and that there is a large degree of redundancy in the sense that the concentration of many molecules detected by GC must be strongly correlated across estates. This conclusion was also supported by the analysis of the concentrations of 32 chemical compounds extracted manually from the chromatogram (Fig. 4), which revealed that estate identity can be significantly decoded using the concentration of any of the 32 compounds in isolation.

The fact that we could perfectly identify estates, independently of vintages, suggests that the estates we have analyzed here have distinct identities. While wine experts believe that some Bordeaux

estates have indeed distinct profiles, this is, to our knowledge the first time that this is demonstrated with a purely chemical analysis of Bordeaux wines. This result was by no means a foregone conclusion. Indeed, one might have worried that GC does not have the required sensitivity for this type of analysis. Ultimately, however, our results reveal that this is not an issue probably because of the redundancy in the chromatograms. As we have seen, many regions can be used to identify estates. Therefore, even if some parts of the chromatograms may not be sensitive enough, the redundancy allows to compensate by looking at other, more sensitive, regions of the chromatogram.

It would be interesting to compare the performance of our model to the one of expert human tasters on blind tasting of the 80 wines we have analyzed. Whether expert wine tasters would be able to match our model's performance (100% correct) on these

seven estates is not known. More generally, it remains to be seen how a GC-based classifier would perform compared to humans on estate, region, or varietal recognition across a wide range of wines. Given our strong performance with estate recognition, artificial and GC-based systems might be able to complement human tasters on wine recognition. We note that other groups have obtained promising results using chemical analysis to predict Rate-All-That-Apply sensory attributes, using techniques that would complement the GC approach presented here[26,27].

To conclude, this study demonstrates that the wine chemical identity of the seven estates considered in this study can be revealed directly from raw chromatograms, without any need for manual extraction of peaks or optimization of the choice of chromatography and ionic scanning. It is quite likely that even better results could be achieved by allowing further tuning of these experimental variables.

## Material and methods

**Wine samples**. Our study involved 80 Bordeaux red wines from 7 estates (a.k.a. chateaux): A, B, C, D, E, F, and G. Twelve vintages were available for each estate (1990, 1995, 1996, 1998, 1999, 2000, 2001, 2002, 2004, 2005, 2006, and 2007) except for estate F which is represented by only eight vintages (1995, 1998, 2000, 2001, 2002, 2005, 2006, and 2007). One single bottle of 75 cL per wine was studied and samples were analyzed in 2018 when the wines were aged between 11 and 28 years in the estates' cellars. A, B, and C are right-bank wines from the Pomerol (A) and St-Emilion (B and C) appellations, while the others are left-bank wines from the Pauillac (D, E), Margaux (F) and Pessac-Léognan (G) appellations.

**Gas Chromatography data acquisition and post-acquisition treatment**. All the wines were analyzed in a single batch in August 2018 by three previously designed GC methods, one focused on the quantification of odorous esters[28], the second one on oak-flavor compounds[29] and the last one on off-flavor compounds[30]. This data set was collected for another set of experiments, unrelated to the current project[24].

*Analysis of off-flavor compounds using SBSE extraction*. The analytical procedure is based on the method developed by Franc et al.[30]. The extraction of compounds was carried out using 10 mm × 1 mm (length × film thickness) polydimethylsiloxane stir bars (Twister®, 63 μL coating, Gerstel, Mülheim an der Ruhr, Germany). These were placed into the vial containing 10 mL of wine sample and 20 μL of an internal standard solution (hydro alcoholic solution 50% v/v at 112 μg L$^{-1}$ of d$_3$-2-isobutyl-3-methoxypyrazine, 55.5 μg.L$^{-1}$ of d$_5$-2,4,6-trichloroanisole 60 μg L$^{-1}$ of d$_5$-4-ethylguaiacol and 300 μg./L$^{-1}$ of (±)-d$_5$-geosmin). After 60 min of stirring at 900 rpm, the stir bars were removed, rinsed with Milli-Q quality water (18.2 MΩ cm$^{-1}$), dried carefully with a paper towel, and transferred into a desorption tube for chromatographic analysis. Analyses were performed using an Agilent 6890 gas chromatograph system, fit with an Agilent HP-5MS capillary column (30 m × 0.25 mm i.d. 0.25 μm film thickness). The GC was combined into an Agilent 5975 mass spectrometric detector (Agilent Technologies, Massy, France). The system was equipped with a Gerstel MPS 2 autosampler, a Twister Thermal Desorption Unit (TDU), and a Gerstel Cooled Injection System Programmable Temperature Vaporization (PTV) inlet (Gerstel, Mülheim an der Ruhr, Germany).The compounds adsorbed on stir bars were thermally desorbed in the TDU with a helium flow at 50 mL min$^{-1}$ in splitless mode and a temperature rate program of 60 °C.min$^{-1}$ from 30 °C to 280 °C (held for 10 min). TDU transfer line was kept at 300 °C and the desorbed compounds were cryofocused in the CIS maintained at −100 °C using liquid nitrogen. PTV injector was

then heated at 12 °C.s$^{-1}$ from −100 °C to 290 °C (held for 5 min) to inject the trapped compounds onto the capillary column. The injection was performed in splitless mode at 1.5 min (helium purge flow of 50 mL.min$^{-1}$). The program for the oven temperature was as follows: 40 °C for initial temperature, increasing to 120 °C at a rate of 2 °C min$^{-1}$ and then to 290 °C (held for 9 min) at a rate of 10 °C min$^{-1}$. Electronic ionization was performed at 70 eV with a detector temperature at 230 °C. The selected ions monitored with a dwell time of 50 ms and distributed in time windows were the following: Window 1—after solvent delay (15 min) to 30.5 min, ions 122, 124, 137, 139, 152, 157; Window 2 - From 30.5 to 34.5 min, ions 195, 197, 210, 212, 215, 217; Window 3 - From 34.5 to 39 min, ions 112, 114, 128, 149, 182, 186; Window 4—From 39 to 43 min, ions 231, 244, 246; Window 5—From 43 to 46 min, ions 265, 278, 280, 329, 331, 346. Run time was 66 min. The method allows the quantification of: 2-Isobutyl-3-methoxypyrazine (IBMP); 2,4,6-trichloroanisole (TCA); 2,4,6-tribromoanisole; 2,3,5,6-tetrachloroanisole; 2,3,4,5,6-pentachloroanisole; 4-ethylphenol (EP); 4-ethylguaiacol (EG); (±)-geosmin. Only, IBMP (in 2 samples at the levels of 4 and 17 ng L$^{-1}$), EP and EG (in all the samples but 3 at levels between 8 and 7000 μg L$^{-1}$) have been detected and quantified.

*Analysis of oak-flavor compounds using liquid/liquid extraction*. Oak-flavour compounds were extracted according to the method described by Bloem[30]: 200 μL of 1-dodecanol (45 mg L$^{-1}$ in hydroalcoholic solution 50% v/v) was added as an internal standard to 50 mL of wine. The sample was extracted three times by 4, 2, and 2 mL of dichloromethane HPLC Grade during 5 min at 700 rpm. The resulting organic extracts were dried with anhydrous sodium sulfate (Na$_2$SO$_4$) and concentrated to 0.25 mL under nitrogen gentle flow. An Agilent 7890 A gas chromatograph coupled with an Agilent 5975 C mass spectrometric detector (Agilent Technologies, Massy, France) and equipped with a Gerstel MPS2 autosampler (Gerstel, Mülheim an der Ruhr, Germany) and a SGE BP21 capillary column (50 m × 0.32 mm i.d., 0.25 μm film thickness) was used (SGE Trajan, Ringwood Victoria, Australia). 2 μL of the extract was injected in splitless mode (30 s) in the injector at 250 °C. The program for the oven temperature was as follows: held 1 min at 60 °C, raised at 4 °C min$^{-1}$ to 220 °C and held for 30 min. MS grade helium was used as carrier gas with a debit of 1 mL min$^{-1}$. Electron ionization was performed at 70 eV with a detector temperature at 180 °C. The selected ions monitored with a dwell time of 20 ms and distributed in time windows were the following: Window 1—after solvent delay (10 min) to 19 min, ions 71, 87, 99, 109, 124; Window 2—From 19 to 28 min, ions 71, 83, 87, 97, 99; Window 3—From 28 to 31 min, ions 111, 131, 139, 149, 154, 164; Window 4—From 31 to 41 min, ions 91, 123, 151, 152, 179, 194; Window 5—From 41 to 43.5 min, ions 123, 151, 166; Window 6—From 43.5 to 50 min, ions 153, 167, 181, 182, 196. Run time was 71 min. The method allows the quantification of: oak-lactone (Whisky-lactone; *cis* and *trans*); eugenol; gaiacol; vanillin; acetovanillone; syringaldehyde; isoeugenol (*cis* and *trans*); furfural; 5-methyl-furfural; acetosynringaldehyde and syringol.

*Analysis of esters using SPME extraction*. We used the method developed by Antalick et al.[28] to obtain a chromatogram that was originally designed to analyze 32 specific esters in wine. In the wines we study here, 16 esters were present at significant concentration, i.e., above the limit of quantification of the method in all the wines[24]. The compounds were extracted by solid-phase micro-extraction (SPME). Ten milliliters of wine and 10 μL of the internal standard solution (mixture of 4,4,4-ethyl-d3-butanoate at 178 mg L$^{-1}$, ethyl-d11-hexanoate at 209 mg L$^{-1}$, ethyl-d15-octanoate at 223 mg L$^{-1}$ and ethyl-d5-*trans*-cinnamate at 325 mg L$^{-1}$ in 100% ethanol v/v)

were added to a vial containing 3.5 g of sodium chloride. SPME fiber (polydimethylsiloxane, 100 μm film thickness, 1 cm length) from Supelco (Bellefonte, Palo Alto, USA) was used as absorbent. Extraction of compounds was performed at 40 °C for 30 min with agitation speed at 500 rpm. Following desorption for 15 min at 250 °C, samples were injected in splitless mode (45 s). Gas chromatographic analysis was carried out on an Agilent 6890 N gas chromatograph coupled to an Agilent 5875 C mass spectrometer (Agilent technologies, Massy, France) and equipped with a Gerstel MPS2 autosampler (Gerstel, Mülheim an der Ruhr, Germany). An SGE BP21 capillary column (50 m × 0.32 mm i.d., 0.25 μm film thickness) was used (SGE Trajan, Ringwood Victoria, Australia), and the carrier gas used was MS grade helium at 1.2 mL min$^{-1}$. The oven temperature was programmed at 40 °C for 5 min then raised at 3 °C min$^{-1}$ to 220 °C (held for 30 min). The mass spectrometer was operated in electron ionization mode at 70 eV with a detector temperature at 280 °C. The selected ions monitored with a dwell time of 20 ms were the following: 43, 55, 56, 57, 60, 61, 69, 70, 71, 74, 85, 87, 88, 89, 91, 97, 99, 101, 102, 104, 105, 106, 110, 114, 116, 122, 127, 131, 136, 142, 176, 178, and 181. They were monitored from after solvent delay (2 min) to 95 min final run time.

*Quantification of specific chemical compounds.* For quantification, the mass spectrometers were operated in selected-ion-monitoring mode. The 32 quantified compounds were identified using retention time associated with control and quantification ions (*m/z*) as described in previous studies[28–30]. The internal standard used to convert area under the peak into concentration was identified using the specific ions as also described in previous studies[28–30]. All the quantifications have been performed using an external calibration set up in an old red Bordeaux wine matrix.

Chromatograms were extracted from Agilent Chemstation software (Agilent Technologies, Santa Clara, CA, USA) as a .csv file without any signal alignment nor baseline correction. These chromatograms specify the signal intensity (A.U.) as a function of retention time. We analyzed the chromatograms separately ("esters", "oak", "offFla") or as a single large chromatogram obtained by concatenating all three chromatograms together along the retention time axis ("concat").

It is important to note that the TIC chromatogram includes peaks produced by the fragmentation of compounds of the wine extract which are not exclusively in the targeted classes. Examples of chromatograms are presented in Fig. S1.

### Data analysis

*Dimensionality reduction.* We used the t-SNE[31] and uniform manifold approximation (UMAP)[32] algorithms to reduce the thousands of dimensions of the chromatograms (corresponding to the retention times) to two (or three dimensions, without any visible structure beyond that seen in two dimensions, Fig. S2), constrained by the distance structure between the chromatograms. The analysis was performed with custom Python scripts, using the scikit-learn library[33] and a Python library for UMAP[32] with default parameters, except perplexity = 30 for tSNE and n_neigbors = 60 for UMAP. Before dimensionality reduction, we pre-processed the data by standard scaling, i.e., for a given chromatogram, each feature (dimension = retention time) was z-scored with respect to all chromatograms (wine samples) of a given type. That means the mean of this feature across samples was subtracted and then it was divided by the standard deviation of this feature across samples (see Fig. S1 for example chromatograms).

*Supervised decoding.* The supervised decoding of estate/vintage from chromatograms was performed with LR and linear discriminant analysis (LDA), implemented in Python, packaged in scikit-learn[33], with default parameters; no hyper-parameter tuning was needed. We used leave-7-out cross-validation for the estate decoding, which consists in splitting the data set into 73 data points for training the classifier and 7 data points for determining generalization performance. The set of seven test wines was obtained by drawing one wine randomly from each of the seven estates. We repeated this split 50 times and reported the average performance over all spits. A similar procedure was used for vintages, but using leave-12-out cross-validation, where 12 is the number of distinct vintages.

*Binning and "survival of the fittest" algorithm.* We tested if sections of the chromatograms contained decodable estate/vintage information by first binning each chromatogram into $N$ non-overlapping consecutive bins of equal size ($N = 50$ in Fig. 3, and $N = 5, 10, 20$ in Fig. S9). For instance, the oak chromatogram has 30 k data points (ordered by retention time), then the first bin contains the first 600 data points, the second the next 600, and so on, and the 50th bin contains the last 600 data points.

We then tested how the iterative removal of bins (sections of the chromatogram) prior to decoding impacted performance, keeping at each step the best bins. More specifically, the procedure is as follows: we started with all $N$ bins and then systematically removed one bin at a time. Each time, we quantified the change in performance when removing one bin. We then selected the bin for which the change in performance was the most favorable (either the smallest drop or largest improvement in performance) and then removed this bin permanently from the chromatogram, resulting in a chromatogram with $N$-1 bins. We repeated this procedure $N$-1 times until only one bin remained and reported performance each time a bin was removed. In other words, this procedure removes iteratively the most informative bins.

We coined this procedure "survival of the fittest" in the sense that the last few bins to be removed are among the most informative ones.

*Statistics.* We used two different one-tailed significance tests. First, decoding estate or vintage resulted in a distribution of decoding accuracies for different train/test splits. Here, a *t* test implemented in Scipy (scipy.stats.ttest_1samp[34]) was used to estimate the likelihood that the chance level is the mean of the distribution of decoding accuracies. We corrected p-values via Bonferroni's method, multiplying them by the number of comparisons (8 for vintage and estate decoding, as there are three chromatogram types, their concatenation, and two classifiers).

## Data availability

The data for this study are available at https://github.com/mschart/wine_decoding.

## Code availability

All the code for the analysis presented here is open source and publicly available at: https://github.com/mschart/wine_decoding All figures can be readily generated from the raw data contained in the same GitHub repository.

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

## Author contributions

A.P., S.M., J.M.B., and M.S. devised the analysis ideas. A.P., S.M., and M.S. wrote the manuscript. M.S. and A.P. performed the analyses. M.S. created figures. J.L., L.R., and S.M. obtained the data and devised the data type.

## Competing interests

The authors declare no competing interests.
