## [Peer Review File · Communications Chemistry]

This manuscript has been previously reviewed at another Nature Portfolio journal. This document only contains reviewer comments and rebuttal letters for versions considered at Communications Chemistry.

Reviewers' comments:

Reviewer #1 (Remarks to the Author):

I have seen previous versions of this manuscript and made substantial comments at that stage in terms of main claims and how convincing they are, novelty, interest to others, conduct of the experiments, discussion in the context of the literature, etc. I pointed out many things that the authors have dealt with already and I really do not have much to add now. The authors are very serious about addressing reviewer feedback and making changes to improve the manuscript, which they have done once again. The work is really well polished now and there is barely anything that needs to be done. One remaining issue is the blending of discussion within the results section. I appreciate the authors' response regarding this providing some context at the point of presenting the results, but that's not the point of having separate Results and Discussion sections. If this approach is to be maintained, then the Results and Discussion sections should be combined, if that's acceptable to the journal. If not, then it would be better not to have any discussion in the results section, despite it being more pleasant to read.

Other than that, some very minor points are:

Introduction: The sentence regarding machine learning and non-linear dimensionality reduction is essentially repeated - it occurs in the 4th paragraph and the 5th, which does not seem necessary.

Figure 1: It could be better to reposition panel c and make it much larger - it could sit in the middle of the overall figure, below panels a and b, above panels d and e. This would not only provide a better ability to see the details in panel c, it would also make the overall figure more symmetrical in appearance.

Mat and Met: Delete the "e" from the end of "geosmin" in the sentences near the end of the SBSE method information.

References: Check for consistent formatting.

Reviewer #4 (Remarks to the Author):

The paper describes a novel approach to classifying wine composition based on GC data. The concept is interesting, the statistical approach is appropriate and the conclusions are generally sound.

The only section that raises some concern in my opinion is the one dealing with the attempt to correlate GC data with journalist's ratings. My concern is mostly linked to two aspects: First, quality evaluations performed by journalist are typically linked to olfactive assessment (orthonasal and retronasal) but also to taste evaluation (eg acidity) and mouthfeel (eg astringency). As GC analyses only addresses volatile compounds, it is not surprising that the correlations observed are weak. Second and more important, the methodology that was adopted for the tasting (as described in the paper) is clearly not appropriate for a scientific context, especially as tastings were not blind (and therefore biased) and variability across replicate tastings is not known. So it appears pointless to try and correlate these two series of data.

My recommendation is to remove this section and retain the rest of the material, which is of high interest for the scientific community.

Reviewer #1 (Remarks to the Author):

I have seen previous versions of this manuscript and made substantial comments at that stage in terms of main claims and how convincing they are, novelty, interest to others, conduct of the experiments, discussion in the context of the literature, etc. I pointed out many things that the authors have dealt with already and I really do not have much to add now. The authors are very serious about addressing reviewer feedback and making changes to improve the manuscript, which they have done once again. The work is really well polished now and there is barely anything that needs to be done.

We thank the reviewer for their constructive feedback which definitely helped us improve the manuscript substantially. Besides a clean, revised version of the manuscript, we also attach a pdf of the old version with tracked changes.

One remaining issue is the blending of discussion within the results section. I appreciate the authors' response regarding this providing some context at the point of presenting the results, but that's not the point of having separate Results and Discussion sections. If this approach is to be maintained, then the Results and Discussion sections should be combined, if that's acceptable to the journal. If not, then it would be better not to have any discussion in the results section, despite it being more pleasant to read.

We are happy to make appropriate changes either way, this is however a decision for the Nature Communications Chemistry editor. We suggest to remove the section title "Discussion" and re-name the results section "Results and Discussion" without any further changes.

Other than that, some very minor points are:

Introduction: The sentence regarding machine learning and non-linear dimensionality reduction is essentially repeated - it occurs in the 4th paragraph and the 5th, which does not seem necessary.

Agreed, we removed "including nonlinear dimensionality reduction" in the 4th paragraph.

Figure 1: It could be better to reposition panel c and make it much larger - it could sit in the middle of the overall figure, below panels a and b, above panels d and e. This would not only provide a better ability to see the details in panel c, it would also make the overall figure more symmetrical in appearance.

We rearranged the panels of figure 1 as suggested, thank you.

Mat and Met: Delete the "e" from the end of "geosmin" in the sentences near the end of the SBSE method information.

Thanks for pointing this out, we corrected the typo.

References: Check for consistent formatting.

We revised the list of references and corrected inconsistencies in formatting (such as authors' names and initials were not consistently formatted, and publication years were not consistently placed within parentheses). Thank you for pointing these out.

Reviewer #4 (Remarks to the Author):

The paper describes a novel approach to classifying wine composition based on GC data. The concept is interesting, the statistical approach is appropriate and the conclusions are generally sound.

We thank the reviewer for this positive assessment. Besides a clean, revised version of the manuscript, we also attached a pdf of the old version with tracked changes.

The only section that raises some concern in my opinion is the one dealing with the attempt to correlate GC data with journalist's ratings. My concern is mostly linked to two aspects: First, quality evaluations performed by journalists are typically linked to olfactory assessment (orthonasal and retronasal) but also to taste evaluation (eg acidity) and mouthfeel (eg astringency). As GC analyses only addresses volatile compounds, it is not surprising that the correlations observed are weak. Second and more important, the methodology that was adopted for the tasting (as described in the paper) is clearly not appropriate for a scientific context, especially as tastings were not blind (and therefore biased) and variability across replicate tastings is not known. So it appears pointless to try and correlate these two series of data.

My recommendation is to remove this section and retain the rest of the material, which is of high interest for the scientific community.

In response to this comment, we removed any section related to Parker's scores predictions from the manuscript. Specifically, we removed the complete section on Parker decoding, two panels in the figure showing our chemical-compounds-based analysis, the Parker-analysis-specific paragraphs in the methods section (decoding details and statistics), the complete supplementary figure showing Parker decoding per GC section, the complete table listing Parker decoding metrics and two panels in the supplementary figure showing example weight distributions for Parker decoding. We also removed the paragraph about Parker decoding in the discussion section and relevant references. All figures and references have been re-numbered accordingly.